# Does the Presence of Circulating Tumor Cells in High-Risk Early Breast Cancer Patients Predict the Site of First Metastasis—Results from the Adjuvant SUCCESS A Trial

**DOI:** 10.3390/cancers14163949

**Published:** 2022-08-16

**Authors:** Elisabeth K. Trapp, Peter A. Fasching, Tanja Fehm, Andreas Schneeweiss, Volkmar Mueller, Nadia Harbeck, Ralf Lorenz, Claudia Schumacher, Georg Heinrich, Fabienne Schochter, Amelie de Gregorio, Marie Tzschaschel, Brigitte Rack, Wolfgang Janni, Thomas W. P. Friedl

**Affiliations:** 1Department of Gynecology and Obstetrics, Medical University of Graz, 8036 Graz, Austria; 2Department of Gynecology and Obstetrics, University Hospital Erlangen, Friedrich-Alexander-University Erlangen-Nuremberg, 91054 Erlangen, Germany; 3Department of Gynecology and Obstetrics, University Hospital Duesseldorf, Heinrich-Heine University Duesseldorf, 40225 Düsseldorf, Germany; 4National Center for Tumor Diseases, Heidelberg University Hospital and German Cancer Research Center, 69120 Heidelberg, Germany; 5Department of Gynecology, University Medical Center Hamburg-Eppendorf, 20246 Hamburg, Germany; 6Breast Center, Department of Gynecology and Obstetrics and CCC Munich, LMU University Hospital, 81337 München, Germany; 7Gynecologic Practice Dr. Lorenz, N. Hecker, Dr. Kreiss-Sender, 38100 Braunschweig, Germany; 8Department of Gynecology and Obstetrics, St. Elisabeth’s Hospital, 50935 Cologne, Germany; 9Oncological Practice, 15517 Fürstenwalde, Germany; 10Department of Gynecology and Obstetrics, University Hospital Ulm, 89081 Ulm, Germany

**Keywords:** liquid biopsy, circulating tumor cells (CTC), breast cancer, metastasis

## Abstract

**Simple Summary:**

Due to recent advances in breast cancer detection and treatment strategies, the number of breast cancer survivors has increased over the past decades. However, breast cancer follow-up guidelines have not changed for years. The presence of CTCs detected during follow-up has been shown to indicate poor prognosis in high-risk breast cancer patients. Here, we evaluated if the presence of CTCs also indicates the site of metastatic disease by analyzing CTC status and metastatic location in 206 patients with distant recurrence from the large adjuvant breast cancer trial SUCCESS A. Patients who were CTC-positive both before and after chemotherapy were more likely to show bone-only first distant disease (37.5% vs. 21.0%) and first distant disease at multiple sites (31.3% vs. 12.6%) than patients without CTCs. These data indicate that CTCs might serve as a liquid biopsy surveillance-marker enabling risk-stratification for deciding on further adjuvant add-on-treatment.

**Abstract:**

The prognostic relevance of circulating tumor cells (CTCs) in breast cancer is well established. However, little is known about the association of CTCs and site of first metastasis. In the SUCCESS A trial, 373 out of 3754 randomized high-risk breast cancer patients developed metastatic disease. CTC status was assessed by the FDA-approved CellSearch^®^-System (Menarini Silicon Biosystems, Bologna, Italy) in 206 of these patients before chemotherapy and additionally in 159 patients after chemotherapy. CTCs were detected in 70 (34.0%) of 206 patients before (median 2 CTCs, 1–827) and in 44 (27.7%) of 159 patients after chemotherapy (median 1 CTC, 1–124); 16 (10.1%) of 159 patients were CTC-positive at both timepoints. The site of first distant disease was bone-only, visceral-only, and other-site-only in 44 (21.4%), 60 (29.1%), and 74 (35.9%) patients, respectively, while 28 (13.6%) patients had multiple sites of first metastatic disease. Patients with CTCs at both timepoints more often showed bone-only first distant disease (37.5% vs. 21.0%) and first distant disease at multiple sites (31.3% vs. 12.6%) than patients without CTCs before and/or after chemotherapy (*p* = 0.027). In conclusion, the presence of CTCs before and after chemotherapy is associated with multiple-site or bone-only first-distant disease and may trigger intensified follow-up and perhaps further treatment.

## 1. Introduction

Breast cancer is the most common cancer in women. Although mortality rates have declined since 1989, recent data indicate that the incidence of breast cancer is rising [1]. While breast cancer remains curable in stages I–III, the main goal of treatment in stage IV remains as extending time with the best possible quality of life. Therefore, metastasis still represents the limit of curable disease.

Furthermore, metastatic disease remains the principal cause of breast-cancer-related death [2,3].

To metastasize, a multi-step process termed metastatic cascade is started at the primary tumor site. In the first step, single tumor cells invade the surrounding tissue and then enter the circulatory and lymphatic system. Therefore, these shedding cells have to escape the immune system and survive surrounded by mesenchymal environment [4].

Epithelial-to-mesenchymal plasticity resembles the key to develop into disseminated tumor cells (DTCs) and circulating tumor cells (CTCs) [5]. These micro-metastatic lesions may enter distant organs like seeds in the soil and stay quiescent in their metastatic niche and/or reactivate, promoting metastasis [6,7].

Several clinical studies have demonstrated the prognostic value of CTC detection before, during, and after completion of chemotherapy both in metastatic and early breast cancer [8,9,10]. Furthermore, it has been shown that positive CTC status during follow-up of high-risk early breast cancer survivors is associated with poor prognosis [11,12].

Over the years, screening-based early breast cancer detection and modern multidisciplinary treatment modalities have improved survival prospects for breast cancer patients constantly [13,14]. Facing the increasing number of breast cancer survivors, follow-up strategies integrating novel prognostic factors to classify breast cancer patients into “risk classes” are lacking [15]. So far, large surveillance trials have failed to demonstrate survival improvements by intensified follow-up [16]. Accordingly, all risk groups are followed up in the same way [17,18,19,20]. However, advances in breast cancer research such as the development of new chemotherapeutics and endocrine or targeted therapies as well as PDL1-, TROP-2, PARP-, and CDK4/6-inhibitors have rendered multiple new and well-tolerated treatment options demonstrating survival benefits in high-risk breast cancer patients. Some of these therapeutics have already made their way into adjuvant add-on treatments for high-risk breast cancer survivors [21,22,23,24,25]. The use of these add-on treatment options is based on the perceived recurrence risk of patients, which is traditionally assessed using clinical data on patients’ age, tumor subtype, tumor size, tumor response to neoadjuvant treatment, lymph node involvement, and pathological grade and type. More recently, risk has been additionally assessed using genomic tests which are intended to complement risk assessment based on traditional clinico-pathological parameters but are only available for certain cancer subtypes. However, data are accumulating that additional markers with better prognostic and predictive value are needed to identify patients who benefit most from additional add-on adjuvant treatment options that enter a clinical routine. As these treatments may cause serious side effects, individual patient risk has to be assessed in the most accurate way, to not only prevent undertreatment, but equally important, to avoid ineffective overtreatment [26,27,28].

Identifying clear high-risk breast cancer patients and offering further adjuvant treatment strategies, therefore, leads to the question of whether early detection and treating micro-metastatic disease might improve survival and may help to prevent symptomatic and overt metastatic disease. Studies regarding treatments targeting DTC and CTC provided conflicting data about the predictive value of micro-metastasis in prospective trial settings [29,30,31,32]. While CTCs showed to be a promising prognostic tool for identifying patients at risk for tumor relapse throughout treatment and follow-up, it is unknown whether the presence of CTCs may also indicate the future site of metastatic lesions. If indeed, the presence of CTCs points not only to a higher risk of relapse but also to a higher likelihood of disease recurrence at specific sites, this may help to decide on new or additional targeted surveillance and/or treatment options, which down the road might prevent or at least delay overt metastatic disease.

## 2. Methods

### 2.1. Study Design

SUCCESS A (NCT 02181101) is a phase III, randomized, prospective, multicenter open label trial with a 2 × 2 factorial design to evaluate the benefit of adding gemcitabine to adjuvant breast cancer chemotherapy and the benefit of extended adjuvant bisphosphonate treatment for five years. The study included 3754 patients who were first randomized to receive either three cycles of epirubicin, fluorouracil, or cyclophosphamide (FEC, 500/100/500 mg/m², q3w) followed by three cycles of docetaxel (Doc, 100 mg/m^2^, q3w) or three cycles of FEC (500/100/500 mg/m^2^, q3w) followed by three cycles of docetaxel and gemcitabine (DocG, docetaxel 75 mg/m^2^ q3w plus gemcitabine 1000 mg/m^2^ d1,8 q3w). After completion of chemotherapy, patients were subject to a second randomization to compare two versus five years of adjuvant zoledronate treatment (4 mg intravenously every 3 months for 2 years versus 4 mg intravenously every 3 months for 2 years followed by 4 mg intravenously every 6 months for 3 years). All trial patients were treated according to national German guidelines regarding endocrine and radiotherapy treatment as well as HER2 targeted therapy, if applicable. More information regarding the study design and the main outcomes with respect to the first and second randomization can be found in the works of De Gregorio et al. (2020) and Friedl et al. [33,34].

The trial was approved by 37 German ethical boards (the main ethics board was Ludwig-Maximilians-University Munich; positive ethics vote 076/05) and conducted between September 2005 (first patient in) and September 2014 (last patient out). The trial was conducted according to Good Clinical Practice guidelines and the Declaration of Helsinki, and all SUCCESS A patients provided written informed consent.

### 2.2. Patients

Eligibility for SUCCESS A required high-risk node-negative (grade 3, negative ER/PR status, age ≤ 35 years or tumor size ≥ pT2) or node-positive invasive breast cancer. Metastatic disease had to be excluded by clear chest X-ray, ultrasound of the abdomen, as well as bone scintigraphy before randomization. The tumor was staged according to the AJCC tumor node metastasis (TNM) system [35] and the histopathological grading [36] was determined in accordance with the Bloom–Richardson system. Receptor status was assessed by immunohistochemistry (IHC). HER2-status was defined positive if membranous staining was strong (IHC 3+) or if immunohistochemical membranous staining was moderate (IHC 2+) and additional HER2-based fluorescence in situ hybridization (FISH) testing was positive. Estrogen- and progesterone-receptor-positivity was assessed by immuno-histochemical nuclear staining, and hormone receptor status was classified as positive if 10% of the investigated cells stained for at least one of the two receptors. All randomized patients underwent surgery with lymph node evaluation, comprising breast-conserving techniques as well as mastectomy, resulting in R0. Metastatic disease was assessed by routine follow-up care according to national German guidelines comprising clinical examinations at 3-month intervals for the first 2 years, every 6 months during the subsequent 3 years, and yearly thereafter, mammography every 12 months, and symptom-driven additional examinations.

### 2.3. CTC Detection

The adjunct translational research program of the SUCCESS A trial included determination of CTC status at four predefined timepoints (before chemotherapy, immediately after chemotherapy, two years after chemotherapy, and five years after chemotherapy). Evaluation of CTC status required written informed consent by the patients to voluntarily participate in this adjunct translational project, which was, however, not mandatory for inclusion in the SUCCESS A study. As some patients refused, CTC status could not be assessed in all SUCCESS A patients. CTC evaluation was performed using the semi-automated FDA-approved CellSearch^®^ system (Menarini Silicon Biosystems, Bologna, Italy; formerly Janssen Diagnostics) as described in detail elsewhere [9,10,37]. Briefly, CTC enumeration was achieved by immunomagnetic enrichment targeting the epithelial cell adhesion molecule (EpCAM). CTCs automatically pre-selected by the CellSearch^®^ system as cytokeratin- and EpCAM-positive nucleated cells lacking CD45 were reviewed by two independent investigators for a final assessment of CTC status. A blood sample was defined as CTC-positive if at least one CTC was detected in 30 mL of peripheral blood.

### 2.4. Statistical Analysis

Descriptive statistics for all the categorical data are summarized with absolute and relative frequencies, while continuous variables are described using median and range. Group comparisons were performed using the Mann–Whitney U test (two groups) or the Kruskal–Wallis test (more than two groups) for continuous variables and the Cochran–Armitage test for trend or the Chi square test for the categorical variables. Associations between the CTC status and site of first metastatic disease were analyzed with the Chi square test.

Patient outcomes were analyzed in terms of distant disease-free survival (DDFS) and overall survival (OS). DDFS was defined as the time from randomization to distant disease recurrence or death from any cause or to the last date of adequate tumor assessment. OS was defined as the time from randomization to death from any cause or to the last date on which the patient was known to be alive. Survival rates based on time-to-event data were estimated by the Kaplan–Meier product limit method and survival curves were presented using Kaplan–Meier survival plots and compared using log-rank tests.

SPSS (IBM, New York, NY, USA) statistical software package, version 24, was used for statistical analysis. All statistical tests were two-tailed and *p* values < 0.05 were considered significant; there was no adjustment of the significance level for multiple comparisons.

## 3. Results

### 3.1. Patient Characteristics

Median follow-up time in the SUCCESS A study was 64 months. In total, 373 (9.9%) of the 3754 SUCCESS A patients developed metastatic disease (median time from randomization until the occurrence of the first metastatic disease within 30.0 months). An amount of 206 (55.2%) of these 373 patients gave their consent for participation in the translational research project and had data on CTC status prior to chemotherapy availability—these 206 patients were included in the analysis. An additional CTC assessment after chemotherapy was available for 159 (77.2%) of the 206 patients.

The median age of the 206 included patients was 55 years (range 27 to 75 years) and the median body mass index was 26.2 kg/m^2^ (range 18.5 to 47.0 kg/m^2^). There were 78 (37.9%) and 128 (62.1%) pre- and postmenopausal patients, respectively. In 114 (55.3%) patients, breast-conserving tumorectomy was performed, while 92 (44.7%) patients underwent mastectomy for achieving R0 status. Overall, 123 (59.7%) patients were hormone-receptor-positive (estrogen receptor only, progesterone receptor only, and both estrogen and progesterone receptor positively stained in 27, 13, and 83 patients, respectively), and 45 (21.8%) patients had an HER2-positive primary tumor. According to the SUCCESS A inclusion criteria demanding high-risk tumors for randomization, most breast cancers were graded poorly differentiated (G3) or moderately differentiated (G2) with 132 (64.1%) and 71 (34.5%) of the cases, respectively. Histological tumor types were invasive ductal in 166 (80.6%), invasive lobular in 29 (14.1%), and other (e.g., medullary or papillary invasive breast cancer) in 11 (5.3%) patients. More details regarding patient and tumor characteristics are given in Table 1. Importantly, the 206 patients with distant recurrence and known CTC status at baseline included in this analysis did not differ significantly from the 167 patients with distant recurrence but unavailable CTC status at baseline with regard to patient or tumor characteristics, with the exception of histological grading (Table 1).

### 3.2. CTC Status before and after Chemotherapy

Of the 206 SUCCESS A patients who developed metastatic disease and for whom CTC status was assessed prior to chemotherapy, 136 (66.0%) were CTC-negative and 70 (34.0%) were CTC-positive. The number of CTCs detected in 30 mL of blood ranged from 1 to 827 CTCs (median 2 CTCs). Of the 159 patients with CTC assessment both before and after chemotherapy, CTC status before chemotherapy was negative for 107 (67.3%) patients and positive for 52 (32.4%) patients (median 2 CTCs, range 1–827 CTCs). After chemotherapy, 115 (72.3%) patients were CTC-negative and 44 (27.7%) patients were CTC-positive (median 1 CTC, range 1–124 CTCs). Combining CTC status prior to and after completion of chemotherapy revealed four different CTC change groups: 79 (49.7%) patients had no CTCs at both timepoints (neg/neg), 28 (17.6%) patients were CTC-negative before but CTC-positive after chemotherapy (neg/pos), 36 (22.6%) patients switched from CTC-positive before chemotherapy to CTC-negative after chemotherapy (pos/neg), and 16 (10.1%) were CTC-positive at both timepoints (pos/pos).

### 3.3. Localisation of First Metastatic Disease

Among the 206 SUCCESS A patients that developed metastatic disease and had a CTC assessment, first metastasis predominantly occurred in visceral sites (lung, liver), accounting for 60 (29.1%) patients. Bone-only first metastatic disease was documented in 44 (21.4%) patients. Other sites of first metastasis (including CNS/brain, lymphogenic sites comprising any lymph node other than supra, -infraclavicular, or axillar or adjacent to the internal mammary artery, pleura, skin, and ovary) were observed in 74 (35.9%) patients. In 28 (13.6%) patients, first metastatic disease was observed in multiple sites at the same time. The distribution of sites of first metastatic lesions (bone only, visceral only, single other site only, multiple sites) did not differ between the 206 SUCCESS A patients with CTC assessment included in the study and the remaining 167 SUCCESS A patients with distant disease without CTC assessment (see Table 1).

### 3.4. Distant Disease-Free Survival (DDFS) and Overall Survival (OS) According to Site of First Metastatic Disease

Figure 1 shows the Kaplan–Meier survival plots for DDFS according to the location of first metastatic disease for the 206 patients included in the study. Median DDFS was 29.3 months for bone only, 37.5 months for visceral only, 35.7 months for other site only, and 29.9 months for multiple sites; however, these differences were not statistically significant (log-rank test, *p* = 0.342).

Overall survival according to the location of first metastatic disease is shown in Figure 2. Patients with first metastatic disease at multiple sites had significantly shorter OS (median 42.0 months) compared to patients with bone-only first metastatic disease (median 68.7 months; log-rank test pairwise comparison: *p* = 0.014), patients with visceral-only first metastatic disease (median 63.6 months; log-rank test pairwise comparison: *p* = 0.016), and patients with first metastatic disease at other sites only (median 67.5 months; log-rank test pairwise comparison: *p* = 0.003). No significant differences with regard to OS were found between patients with bone-only, visceral only, and other site only first metastatic disease (log-rank test pairwise comparisons: all *p* > 0.40).

Similar results for DDFS and OS were obtained when only the 159 patients with CTC assessments both before and after chemotherapy were analyzed. DDFS did not differ significantly among the locations of first metastatic disease (bone only, visceral only, single other site only, multiple sites; log-rank test, *p* = 0.259), but OS did show significant differences among the four locations of first metastatic disease (log-rank test, *p* = 0.022). Patients with first metastatic disease at multiple sites had significantly shorter OS compared to each of the other metastatic location groups (log-rank test pairwise comparisons: all *p* < 0.04), while no significant differences with regard to OS were found between patients with bone-only, visceral only, and other site only first metastatic disease (log-rank test pairwise comparisons: all *p* > 0.40).

### 3.5. CTC Status and Metastatic Site

Compared to patients who were CTC-negative before chemotherapy, CTC-positive patients were numerically more likely to have bone-only first distant disease (25.7% vs. 19.1%) and to have more than one metastatic site at the time first distant recurrence was diagnosed (20.0% vs. 10.3%); the differences between CTC-positive and CTC-negative patients regarding the location of first distant recurrence (bone-only, visceral-only, other site only, multiple sites) were, however, not significant (Chi-squared test, *p* = 0.087; Table 2). Similar proportions were observed with regard to metastatic site at the time of first distant recurrence when only the 159 patients with CTC assessment both before and after chemotherapy were analyzed. CTC-positive patients were numerically more likely to have bone-only first distant disease (23.1% vs. 22.4%) and to have more than one metastatic site at the time first distant recurrence was diagnosed (25.0% vs. 9.3%); however, the differences between CTC-positive and CTC-negative patients regarding the location of first distant recurrence (bone-only, visceral-only, other site only, multiple sites) were again not significant by a small margin (Chi-squared test, *p* = 0.056; Table 2). Accordingly, the number of CTCs detected before chemotherapy also did not differ significantly among these four metastatic location groups, both when all 206 patients were analyzed (Kruskal–Wallis test, *p* = 0.097) and when only the 159 patients with CTC assessment before and after chemotherapy were analyzed (Kruskal–Wallis test, *p* = 0.068).

For the 159 patients with CTC assessment after chemotherapy, no association between the site of first metastatic disease and CTC status after chemotherapy was found (Chi-squared test, *p* = 0.218; Table 2) and the number of CTCs detected after chemotherapy also did not differ significantly among these four groups (Kruskal–Wallis test, *p* = 0.194).

A comparison of the sites of first metastatic disease among all four CTC change groups revealed differences of borderline significance (Chi-squared test, *p* = 0.074; Table 3). When we compared the CTC pos/pos group with the other three CTC change groups combined, these differences became significant (Chi-squared test, *p* = 0.027), with patients who were CTC-positive at both timepoints more frequently presenting with bone-only first distant disease (37.5% vs. 21.0%) and first distant disease at multiple sites (31.3% vs. 12.6%) compared to patients who were not CTC-positive at both timepoints.

## 4. Discussion

In the present study, we were able to show that the presence of CTCs detected both before and after adjuvant chemotherapy treatment in peripheral blood samples of early breast cancer patients who subsequently developed distant recurrences indicate an increased risk for bone metastases and multi-metastatic disease. Patients with persistently positive CTC status more frequently showed osseous or multiple metastatic disease as compared to patients with discordant or persistently negative CTC status. Patients with first metastatic disease at multiple sites showed shorter survival compared to patients with first distant recurrence at one site only.

The presence of CTCs before and after chemotherapy is associated with poor prognosis and higher risk for relapse, as these CTCs are interpreted to represent chemotherapy-resistant “minimal residual disease” [9,10]. Furthermore, it could be shown that the presence of CTCs during follow-up, two years after the completion of primary breast cancer treatment, was associated with reduced disease-free and overall survival, especially in patients who demonstrated positive CTC status both before treatment and during follow-up [12]. These results are in line with the concept of the metastatic cascade, which postulates that tumor dormancy, epithelial-to-mesenchymal plasticity, or stemness protect minimal distant residual disease from systemic treatment [4,5,7,38].

As we are able to detect disseminated tumor cells in the bone marrow even before overt metastasis is present, the idea of CTCs and DTCs predicting future metastatic disease, especially in the bones, appears obvious [7,9,39]. Bone not only is a preeminent site for metastasis in the breast but also in prostate cancer and is affected in about 65% to 70% of metastatic patients. Bone is a continuously remodeling organ and the homeostasis between osteoblasts and osteoclasts is governed by systemic estrogen. Metastatic outgrowth is based on aberrant bone remodeling achieved by paracrine crosstalk between DTCs, osteoclasts, and osteoblasts, as well as the bone matrix [40].

Accordingly, data from several large prospective trials suggest that antiresorptive bone treatment with bisphosphonates not only treats bone loss associated with endocrine therapy but also prolongs distant disease-free survival and reduces breast cancer mortality [34,41,42]. In line with these findings, DTC-positive patients treated with bisphosphonates demonstrated a reduction in bone marrow DTCs over time [43,44]. In a stage II/III high-risk breast cancer population treated with adjuvant zoledronate, Neelima et al. recently observed a decline in DTCs and also in CTCs. However, data especially regarding CTC clearance by adjuvant bisphosphonates remain controversial and prospective randomized trials are lacking [31,44]. In the SUCCESS A study, all patients were randomized to receive either 2 or 5 years of adjuvant zoledronic acid treatment (see methods). However, CTC positivity assessed after 5 years did not differ significantly between patients with 2 or 5 years of zoledronate treatment. In addition, the prolonged adjuvant treatment with 5 vs. 2 years of zoledronic acid did not significantly improve disease-free survival, overall survival, and distant disease-free survival [34].

In our analysis, CTC positivity before and after chemotherapy was associated not only with osseous but also with multi-site first metastatic disease, which by itself indicated worse overall survival compared to single-site first distant disease. Multi-metastatic recurrence as first metastatic disease may be explained by tumor heterogeneity and the process of epithelial-mesenchymal transition (EMT). Current studies analyzing single CTCs suggest that they are heterogenous even when harvested from the same individual and therefore belong to different neoplastic subpopulations [3]. Furthermore, they undergo EMT to survive the many bottlenecks on the way to a premetastatic niche, such as the bone marrow. By EMT, these cells lose their epithelial characteristics which makes them more prone to metastasize [38,45]. Especially, this stem-cell-like subpopulation of EMT-affected and metastasis-driving CTCs is missed by the CellSearch system, which detects CTCs according to their epithelial cell adhesion molecule (EpCAM) expression [46].

However, this does not exclude the possibility that EpCAM-positive CTCs form metastatic lesions. Baccelli et al. were able to generate metastatic disease in immunodeficient mice by CellSearch-evaluated CTC-injections. Six recipient rodents developed multiple, visceral, and bone metastases within 6 to 12 months after CTC-injection derived from metastatic breast cancer patients. They concluded that there are different subgroups of CTCs and only a small proportion of cells, so-called metastasis-initiating cells (MIC), has the potential to cause multi-metastatic disease. In line with our clinical data, these MIC originated metastatic lesions occurred in multiple organ systems simultaneously [47].

Our analysis has some limitations. Although the SUCCESS A study was a large trial with 3754 high-risk primary breast cancer patients being recruited, the analysis presented here is based on a small subset of 206 patients, who developed metastatic disease and participated in the translational research project. CTC status at both time points could be assessed in only 159 patients. However, the patients included here were similar with respect to clinico-pathological factors compared to patients with distant disease recurrence that did not participate in the translational research project with CTC assessments; thus, no bias is expected with regard to the associations between CTC status and the site of first metastatic disease found in our study. Due to the small sample size, additional subgroup analysis (e.g., regarding intrinsic tumor subtypes) could not be performed. Furthermore, the group of patients with a positive CTC result both before and after chemotherapy consisted of only 16 patients. On the other hand, these data were derived from a large randomized prospective multicenter quality-controlled clinical trial with a long follow-up period of 64 months. Clearly, more data are needed to confirm our results of an association between CTC status and the site of metastatic disease.

While the independent prognostic potential of CTC testing during treatment and follow-up in breast cancer patients at risk for metastatic disease has been confirmed in several studies [9,11,12], current guidelines counsel against intensified diagnostic testing for metastasis in asymptomatic breast cancer survivors [48,49]. However, taking into account the recent progress in developing new adjuvant add-on treatments, CTC enumeration, profiling, and further characterization may provide important information regarding potential therapeutic targets [46,50].

The clinical use of CTC characterization was assessed in the DETECT study program in a metastatic setting [51]. The screening of 1933 HER2-negative metastatic breast cancer patients for HER2-positive CTCs revealed that 174 (15%) patients out of 1159 CTC-positive patients showed HER2-positive CTCs. In the era of new potent agents like Trastuzumab-Deruxtecan, such CTC characterization can be an important step forward for CTCs to translate from purely prognostic into predictive relevance [52].

CTCs occurrence in peripheral blood samples of early breast cancer patients resembles a low frequency event [53], somewhat limiting the suitability of CTCs as a sole liquid biopsy marker for the detection of minimal residual disease (MRD) during follow-up in early breast cancer survivors. Combining CTC-enumeration with the detection of circulating tumor DNA (ctDNA), shed by tumors during apoptosis, might overcome this problem and considerably increase sensitivity. The prognostic relevance of ctDNA in early breast cancer has been demonstrated in several studies [54,55]. Furthermore, the detection of molecular relapse during breast cancer follow-up via personalized ctDNA profiling has generated a lead-time preceding the clinical detection of metastasis for up to two years. Finally, ctDNA profiling can even identify therapeutic targets, as it just has been established for treatment with alpelisib in ctDNA-PIK3CA mutant metastatic breast cancer patients [56,57].

Whether liquid biopsy is suitable for monitoring and surveillance of breast cancer patients will be assessed in the follow-up trial SURVIVE. The SURVIVE study is a large randomized phase III breast cancer surveillance trial that combines these approaches by using different liquid biopsy markers (CTCs, ctDNA, conventional tumor markers) to evaluate whether an intensified breast cancer follow-up with repeated blood sampling for MRD detection and subsequent early treatment intervention leads to an overall survival benefit as compared to standard follow-up care according to current guidelines. The SURVIVE trial, funded by the German Federal Ministry of Education and Research (BMBF), will start in late 2022 and, if successful, could pave the way for liquid biopsy to enter clinical practice and contribute to a paradigm shift in the current follow-up care of medium- and high-risk early breast cancer survivors.

## 5. Conclusions

A positive CTC status before and after adjuvant chemotherapy in early breast cancer patients is associated with an enhanced risk of osseous metastatic disease or distant disease recurrence at multiple sites simultaneously. Thus, these patients may be candidates for additional therapeutic interventions and—given the unfavorable prognostic value of first distant disease recurrence at multiple sites—may also benefit from intensified follow-up surveillance to allow early treatment intervention.

## Figures and Tables

**Figure 1 cancers-14-03949-f001:**
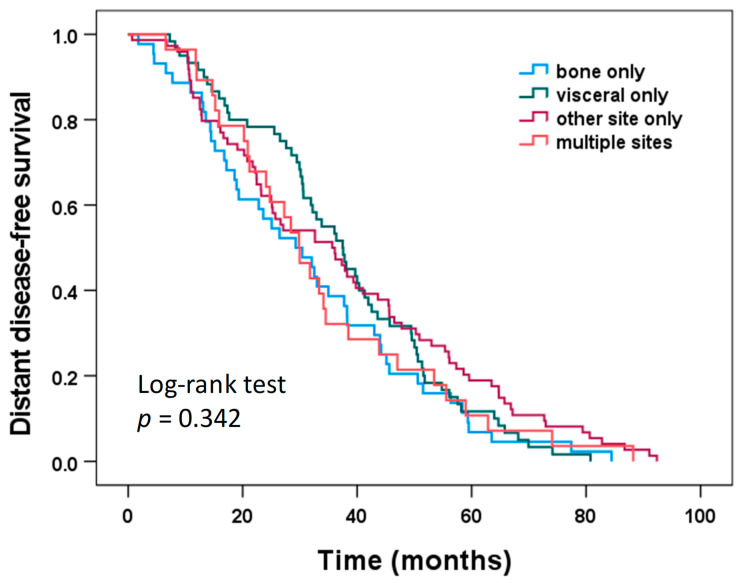
Distant disease-free survival according to the site of first metastatic disease.

**Figure 2 cancers-14-03949-f002:**
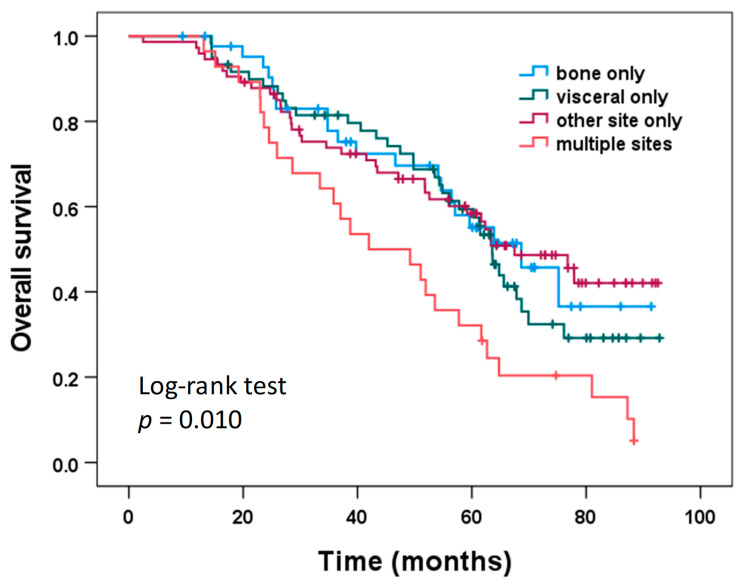
Overall survival according to the site of first metastatic disease.

**Table 1 cancers-14-03949-t001:** Baseline characteristics and clinicopathological variables of patients with early breast cancer recruited in the SUCCESS A trial who developed distant disease according to CTC assessment at baseline before the start of adjuvant chemotherapy (yes/no).

	CTC Assessment at Baseline		
Variable	Yes *n* = 206	No *n* = 167	*p*-Value ^1^
Age (years)			0.279 ^2^
*Median*	55	54	
*Range*	27–75	26–86	
Body mass index (kg/m^2^)			0.420 ^2^
*Median*	26.2	25.9	
*Range*	18.5–47.0	16.7–44.9	
Tumor stage			0.463 ^3^
*pT1*	57 (27.7%)	49 (29.3%)	
*pT2*	121 (58.7%)	100 (59.9%)	
*pT3*	20 (9.7%)	13 (7.8%)	
*pT4*	8 (3.9%)	5 (3.0%)	
Nodal stage			0.693 ^3^
*pN0*	44 (21.4%)	36 (21.6%)	
*pN1*	79 (38.3%)	60 (35.9%)	
*pN2*	42 (20.4%)	34 (20.4%)	
*pN3*	41 (19.9%)	37 (22.2%)	
Histological grading			0.042 ^3^
*G1*	3 (1.5%)	3 (1.8%)	
*G2*	71 (34.5%)	75 (44.9%)	
*G3*	132 (64.1%)	89 (53.3%)	
Histological type			0.411 ^4^
*ductal*	166 (80.6%)	141 (84.4%)	
*lobular*	29 (14.1%)	16 (9.6%)	
*other*	11 (5.3%)	10 (6.0%)	
Hormone receptor status			0.112 ^4^
*negative*	83 (40.3%)	81 (48.5%)	
*positive*	123 (59.7%)	86 (51.5%)	
HER2 status			0.332 ^4^
*negative*	158 (76.7%)	122 (73.1%)	
*positive*	45 (21.8%)	44 (26.3%)	
*unknown*	3 (1.5%)	1 (0.6%)	
Menopausal status			0.497 ^4^
*premenopausal*	78 (37.9%)	69 (41.3%)	
*postmenopausal*	128 (62.1%)	98 (58.7%)	
Type of surgery			0.678 ^4^
*breast conserving*	114 (55.3%)	96 (57.5%)	
*mastectomy*	92 (44.7%)	71 (42.5%)	
Adjuvant chemotherapy arm ^5^			0.724 ^4^
*FEC-DocG*	100 (48.5%)	78 (46.7%)	
*FEC-Doc*	106 (51.5%)	89 (53.3%)	
Zoledronate treatment arm			0.217 ^4^
*5 years*	107 (51.9%)	76 (45.5%)	
*2 years*	99 (48.1%)	91 (54.5%)	
Radiotherapy			0.600 ^4^
*no*	20 (9.7%)	19 (11.4%)	
*yes*	186 (90.3%)	148 (88.6%)	
Endocrine therapy			0.2214
*no*	70 (34.0%)	67 (40.1%)	
*yes*	136 (66.0%)	100 (59.9%)	
HER2-targeted therapy			0.490 ^4^
*no*	164 (79.6%)	128 (76.6%)	
*yes*	42 (20.4%)	39 (23.4%)	
Site of first metastatic disease			0.260 ^4^
*Bone only*	44 (21.4%	33 (19.8%)	
*Visceral only*	60 (29.1%)	37 (22.2%)	
*Single other site only*	74 (35.9%)	76 (45.5%)	
*Multiple sites*	28 (13.6%)	21 (12.6%)	

^1^ All tests without unknowns; ^2^ Mann–Whitney U test; ^3^ Cochran–Armitage test for trend; ^4^ Chi-square test; ^5^ FEC-DocG: 3 cycles of fluorouracil-epirubicin-cyclophosphamide followed by 3 cycles of docetaxel and gemcitabine; FEC-Doc: 3 cycles of fluorouracil-epirubicin-cyclophosphamide followed by 3 cycles of docetaxel.

**Table 2 cancers-14-03949-t002:** Association between site of first metastatic disease recurrence and both CTC status at baseline (i.e. before chemotherapy) and CTC status after chemotherapy (*n* = 206: all patients with CTC assessment at baseline, i.e., before chemotherapy; *n* = 159: only patients with CTC assessment both before and after chemotherapy).

Location of First Distant Recurrence	CTC Status at Baseline (*n* = 206)		CTC Status at Baseline (*n* = 159)		CTC Status after Chemotherapy (*n* = 159)	
	CTC Negative (*n* = 136)	CTC Positive (*n* = 70)	CTC Negative (*n* = 107)	CTC Positive (*n* = 52)	CTC Negative (*n* = 115)	CTC Positive (*n* = 44)
Bone only	26 (19.1%)	18 (25.7%)	24 (22.4%)	12 (23.1%)	23 (20.0%)	13 (29.5%)
Visceral only	45 (33.1%)	15 (21.4%)	36 (33.6%)	12 (23.1%)	37 (32.2%)	11 (25.0%)
Other site only	51 (37.5%)	23 (32.9%)	37 (34.6%)	15 (28.8%)	41 (35.7%)	11 (25.0%)
Multiple sites	14 (10.3%)	14 (20.0%)	10 (9.3%)	13 (25.0)	14 (12.2%)	9 (20.5%)

**Table 3 cancers-14-03949-t003:** Association between site of first metastatic disease recurrence and CTC change group.

Location of First Distant Recurrence	CTC Change Group (*n* = 159)			
	neg/neg (*n* = 79)	neg/pos (*n* = 28)	pos/neg (*n* = 36)	pos/pos (*n* = 16)
Bone only	17 (21.5%)	7 (25.0%)	6 (16.7%)	6 (37.5%)
Visceral only	29 (36.7%)	7 (25.0%)	8 (22.2%)	4 (25.0%)
Other site only	27 (34.2%)	10 (35.7%)	14 (38.9%)	1 (6.3%)
Multiple sites	6 (7.6%)	4 (14.3%)	8 (22.2%)	5 (31.3%)

## Data Availability

The data presented in this study are available on request from the corresponding author.

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
