# Peer review of "Does the Presence of Circulating Tumor Cells in High-Risk Early Breast Cancer Patients Predict the Site of First Metastasis—Results from the Adjuvant SUCCESS A Trial"

_cancers, 2022, doi:10.3390/cancers14163949_

Round 1

Reviewer 1 Report

The prognostic relevance of circulating tumor cells (CTCs) in breast cancer is well established. However, little is known about the association of CTCs and site of first metastasis. 3754 high-risk breast cancer patients were randomized in the SUCCESS A trial. Overall, 373 patients metastasized, and in 206 of these patients CTC-status before chemotherapy was assessed by the FDA-approved CellSearch®-System (Menarini Silicon Biosystems). 1. CTCs were determined before and after chemotherapy, however, data on the content of CTCs are available only for 159 patients, so I consider it appropriate to include these patients in the study, since the authors discuss exactly both statuses (before and after chemotherapy). 2. If the groups are equal, then the % in table 2 may change, this factor must be taken into account. Without recalculation for 159 people before the start of chemotherapy, I do not consider it appropriate to publish the results of the study.

Author Response

Thank you for your overall supportive comment.

Point 1: We are grateful to Reviewer 1 for this important comment. We fully agree that it is important to demonstrate that results obtained for the larger cohort with 206 patients that have a CTC assessment only before chemotherapy are also valid for the reduced cohort of 159 patients with CTC assessments at both timepoints. Accordingly, we have added the following results obtained for the 159 patient cohort:

  • Number of CTCs detected before chemotherapy:
    “Of the 159 patients with a CTC assessment both before and after chemotherapy, CTC status before chemotherapy was negative for 107 (67.3%) patients and positive for 52 (32.4%) patients (median 2 CTCs, range 1 - 827 CTCs).”
  • Comparison of distant disease-free survival (DDFS) and overall survival (OS) according to the locations of first metastatic disease (bone only, visceral only, single other site only, multiple sites):
    “Similar results for DDFS and OS were obtained when only the 159 patients with CTC assessments both before and after chemotherapy were analyzed. DDFS did not differ significantly among the locations of first metastatic disease (bone only, visceral only, single other site only, multiple sites; log-rank test, p = 0.259), but OS did show significant differences among the four locations of first metastatic disease (log-rank test, p = 0.022). Patients with first metastatic disease at multiple sites had significantly shorter OS compared to each of the other metastatic location groups (log-rank test pairwise comparisons: all p < 0.04), while no significant differences with regard to OS were found between patients with bone-only, visceral only and other site only first metastatic disease (log-rank test pairwise comparisons: all p > 0.40).”
  • Comparison of locations of metastatic sites between patients that were CTC-positive or CTC-negative before chemotherapy:
    “Similar proportions were observed with regard to metastatic site at the time first distant recurrence when only the 159 patients with CTC assessment both before and after chemotherapy were analyzed. CTC-positive patients were numerically more likely to have bone-only first distant disease (23.1% vs 22.4%) and to have more than one metastatic site at the time first distant recurrence was diagnosed (25.0% vs 9.3%); however, the differences between CTC-positive and CTC-negative patients regarding the location of first distant recurrence (bone-only, visceral-only, other site only, multiple sites) were again not significant by a small margin (Chi-squared test, p = 0.056; Table 2). Accordingly, the number of CTCs detected before chemotherapy also did not differ significantly among these four metastatic location groups, both when all 206 patients were analyzed (Kruskal Wallis test, p = 0.097) and when only the 159 patients with CTC assessment before and after chemotherapy were analyzed (Kruskal Wallis test, p = 0.068).”

Thus, the results obtained for the reduced cohort of 159 patients with CTC assessments at both timepoints are more or less the same than the results of the larger cohort of 206 patients.

While we fully agree with Reviewer 1 that the addition of these results is crucial and important, we do not see the need for omitting the results for the analyses based on the larger sample size of 206 patients altogether, as these results complement and confirm rather than contradict the results obtained for the reduced cohort of 159 patients. Furthermore, we think that these data help to highlight the significance and added value of multipe CTC-assessments.

Thus, we would prefer to keep the results for the larger cohort of 206 patients as part of the manuscript, and would be happy if editor and reviewers agree with us.

Point 2: We thank Reviewer 1 for the valuable suggestion to edit Table 2 by including data on the association between site of first metastatic disease recurrence and CTC status at baseline as recalculated for the cohort of 159 patients with CTC assessment both before and after chemotherapy.

We followed this advice and have added an additional column with the requested data to Table 2 (see middle column in the revised Table 2 with the header “CTC status at baseline (n = 159)”); these data are now together with the results of the statistical tests also presented in the text (see our response to Point 1 of Reviewer 1 above.

Reviewer 2 Report

This study is based on 206 patients with distant recurrence from the large adjuvant breast cancer trial SUCCESS A. Through data analysis of their CTCs status and metastatic sites, it was found that a positive CTC-status before and after adjuvant chemotherapy in early breast cancer patients is associated with an enhanced risk of osseous metastatic disease or distant disease recurrence at multiple sites simultaneously. This work provides a reference for evaluating the prognosis of breast cancer patients and early treatment intervention.

Strengths:

The article is logical , the data analysis is detailed and reliable, and has certain significance for the prognosis of cancer patients.

Suggestion:

Before the article is accepted, it is suggested to add more status of breast cancer in the introduction part, and the importance of doing a good job in evaluating the prognosis of patients, so as to make the content of the article richer and more complete.

Author Response

Response 2:

Point 1: We are honored for your comment about the strengh of the manuscript.

Point 2: Thank you for the valuable suggestion, underlining the importance of implementing measures to properly evaluate prognosis and risk of patients. We have added a paragraph addressing the importance to assess prognosis and clinical risk in early breast cancer patients, especially in the context of treatment decision. We pointed out that improved risk assessment might help to prevent not only under, but also overtreatment:

“… These add- on treatment options are based on traditional risk stratifying-assessment, like tumor size, tumor response to neoadjuvant treatment, lymph node involvement, pathological grade and type, cancer activity. Although clinical risk has been assessed by traditional makers for years, data are accumulating, that additional markers are needed to rule out who benefits from systemic treatment especially now, that additional add on adjuvant treatment options enter clinical routine. These treatments may cause serious side effects, demanding individual patient risk has to be assessed in the most accurate way, to not only prevent undertreatment, but more importantly ineffective overtreatment.[23][25]

Reviewer 3 Report

The manuscript by Elisabeth K. Trapp et al. Does the presence of circulating tumor cells in high-risk early breast cancer patients predict the site of first metastasis - results from the adjuvant SUCCESS A trial. The paper have good a mechanistic approach, but it provides the data that can be the basis for further research regarding mechanisms. Therefore, it can be of interest to oncologist and biologists. My detailed comments and suggestions are as follows:

1) Abstract - please rearrange the sentence in the text because it's not clear.

2) Indroduction is too short.

3). The disscusion should be more detailed. 

Author Response

Response 3:

Point 1: Unfortunately it is unclear which sentence in the abstract Reviewer 3 is referring to here. However, we have slightly modified and restructured the abstract and hope that it is clearer now.

Point 2: We are grateful to Reviewer 3 for this comment! Following this suggestion (and also based on comments from Reviewer 2) we added a paragraph about prognosis and risk-assessment to tailor anti-tumor treatment to the introduction:

 Breast cancer is the leading cancer in women. Although mortality rates declined since 1989, recent statistical data indicate that the incidence for breast cancer is rising.[1] While breast cancer remains curable in Stage I-III, the main goal of treatment in stage IV remains extending time with the maximum of quality in life. Therefore, metastasis still represents the limit of curable disease.

and

These add- on treatment options are based on traditional risk stratifying-assessment, like tumor size, tumor response to neoadjuvant treatment, lymph node involvement, pathological grade and type, cancer activity. Although clinical risk has been assessed by traditional makers for years, data are accumulating, that additional markers are needed to rule out who benefits from systemic treatment especially now, that additional add on adjuvant treatment options enter clinical routine. These treatments may cause serious side effects, demanding individual patient risk has to be assessed in the most accurate way, to not only prevent undertreatment, but more importantly ineffective overtreatment.[23][25]

Point 3) We are grateful for your suggestion and have extended the discussion focussing especially on the potential of CTCs and other liquid biopsy markers to provide not only prognostic but also predictive information and to detect minimal residual disease during follow-up. Furthermore, we expanded the sections on the the use of bisphosphonates and the osseous metastatic cascade.

Round 2

Reviewer 1 Report

I have no more comments on the article.